# Credal Marginal MAP

**Radu Marinescu**
IBM Research, Ireland
radu.marinescu@ie.ibm.com

**Debarun Bhattacharjya**
IBM Research, USA
debarunb@us.ibm.com

**Junkyu Lee**
IBM Research, USA
junkyu.lee@ibm.com

**Alexander Gray**
IBM Research, USA
alexander.gray@ibm.com

**Fabio Cozman**
Universidade de São Paulo, Brazil
fgcozman@usp.br

## Abstract

Credal networks extend Bayesian networks to allow for imprecision in probability values. Marginal MAP is a widely applicable mixed inference task that identifies the most likely assignment for a subset of variables (called MAP variables). However, the task is extremely difficult to solve in credal networks particularly because the evaluation of each complete MAP assignment involves exact likelihood computations (combinatorial sums) over the vertices of a complex joint credal set representing the space of all possible marginal distributions of the MAP variables. In this paper, we explore Credal Marginal MAP inference and develop new exact methods based on variable elimination and depth-first search as well as several approximation schemes based on the mini-bucket partitioning and stochastic local search. An extensive empirical evaluation demonstrates the effectiveness of our new methods on random as well as real-world benchmark problems.

## 1 Introduction

Probabilistic graphical models such as Bayesian networks [1] provide a powerful framework for reasoning about conditional (in)dependency structures over many variables. Marginal MAP (MMAP) is the task that identifies the most likely instantiation for a subset of variables given some evidence in a Bayesian network. Since MMAP inference distinguishes between maximization variables (called MAP variables) and summation variables, it is computationally more difficult than either max- or sum-inference tasks alone, primarily because summation and maximization operations do not commute; this forces processing along constrained variable orderings that may have significantly higher induced widths [2, 3, 4]. MMAP is NP$^{PP}$-complete, but despite its complexity, it is often the appropriate task for problems that involve hidden variables such as conformant planning [5], image segmentation with hidden fields [6], or probabilistic diagnosis in healthcare [7].

In many practical situations it may not always be possible to provide the precise specification of a Bayesian network's parameters (i.e., probability values). Credal networks [8] provide an elegant extension to Bayesian networks that retain the graphical appeal of the latter while allowing for a more flexible quantification of the probability values via credal sets. This ability to represent and reason with imprecision in probability values allows credal networks to often reach more conservative and robust conclusions than Bayesian networks. Over the past three decades, the bulk of research has

focused on developing marginal inference algorithms that are concerned with computing efficiently the marginal probability of a query variable given some evidence in the credal network [9].

Abductive reasoning tasks such as explaining the evidence in a credal network with or without hidden (unobserved) variables (i.e., Marginal MAP inference) are equally important to consider in practice. For example, in a hypothetical medical diagnosis situation modeled as a credal network, one may be interested in identifying the most likely combination of underlying medical conditions that determine a negative CT scan result and the presence of severe memory loss [10, 11]. Furthermore, since probabilistic structural causal models can be mapped exactly into equivalent credal networks, credal Marginal MAP inference could also be used to enable effective counterfactual analysis [12].

**Contributions** In this paper, we address the Marginal MAP inference task in credal networks. Specifically, we define the Credal Marginal MAP (CMMAP) task as finding an assignment to a subset of variables that has maximum *upper* (respectively, *lower*) *marginal probability*. We focus first on exact inference and propose a variable elimination as well as a depth-first search scheme for CMMAP. The complexity analysis of these exact methods indicates that they are likely to be limited to very easy problems. Therefore, we subsequently propose a mini-bucket partitioning based approximation of variable elimination for CMMAP as well as a family of approximate search based schemes that combine stochastic local search algorithms such as hill climbing, taboo search and simulated annealing with approximate maginal inference for credal networks. We evaluate empirically the new CMMAP inference algorithms on random credal networks with different graph topologies as well as a collection of credal networks derived from real-world applications. Our experimental results show that the exact approaches are limited to solving relatively small scale problems, while the approximation schemes can scale to much larger and practical problems.

The supplementary material includes additional details, experimental results, code and benchmarks.

## 2 Background

### 2.1 Bayesian Networks

A *Bayesian network* (BN) [1] is defined by a tuple $\langle \mathbf{X}, \mathbf{D}, \mathbf{P}, G \rangle$, where $\mathbf{X} = \{X_1, \ldots, X_n\}$ is a set of variables over multi-valued domains $\mathbf{D} = \{D_1, \ldots, D_n\}$, $G$ is a directed acyclic graph (DAG) over $\mathbf{X}$ as nodes where each $X_i$ has a set of parents $\Pi_i$, and $\mathbf{P}$ is a set of *conditional probability tables* (CPTs) where each $P_i = P(X_i|\Pi_i)$). A Bayesian network represents a joint probability distribution over $\mathbf{X}$, namely $P(\mathbf{X}) = \prod_{i=1}^{n} P(X_i|\Pi_i)$.

Let $\mathbf{X}_M = \{X_1, \ldots, X_m\}$ be a subset of $\mathbf{X}$ called MAP variables and $\mathbf{X}_S = \mathbf{X} \setminus \mathbf{X}_M$ be the complement of $\mathbf{X}_M$, called sum variables. The Marginal MAP (MMAP) task seeks an assignment $\mathbf{x}_M^*$ to variables $\mathbf{X}_M$ having maximum probability. This requires access to the marginal distribution over $\mathbf{X}_M$, which is obtained by summing out variables $\mathbf{X}_S$:

$$\mathbf{x}_M^* = \underset{\mathbf{X}_M}{\operatorname{argmax}} \sum_{\mathbf{X}_S} \prod_{i=1}^{n} P(X_i|\Pi_i) \tag{1}$$

MMAP is a mixed inference task (max-sum) and its complexity is known to be $\mathrm{NP^{PP}}$-complete [3]. Over the past decades, several algorithmic schemes have been developed for solving MMAP efficiently. We later overview the most relevant exact and approximate algorithms for MMAP.

### 2.2 Credal Networks

A set of probability distributions for variable $X$ is called a *credal set* and is denoted by $K(X)$ [13]. Similarly, a *conditional credal set* is a set of conditional distributions, obtained by applying Bayes rule to each distribution in a credal set of joint distributions [14]. We consider credal sets that are closed and convex with a finite number of vertices. Two credal sets $K(X|Y = y_1)$ and $K(X|Y = y_2)$, where $y_1$ and $y_2$ are two distinct values of variable $Y$, are called *separately specified* if there is no constraint on the first set that is based on the properties of the second set.

A *credal network* (CN) [8] is defined by a tuple $\langle \mathbf{X}, \mathbf{D}, \mathbf{K}, G \rangle$, where $\mathbf{X} = \{X_1, \ldots, X_n\}$ is a set of discrete variables with finite domains $\mathbf{D} = \{D_1, \ldots, D_n\}$, $G$ is a directed acyclic graph (DAG) over

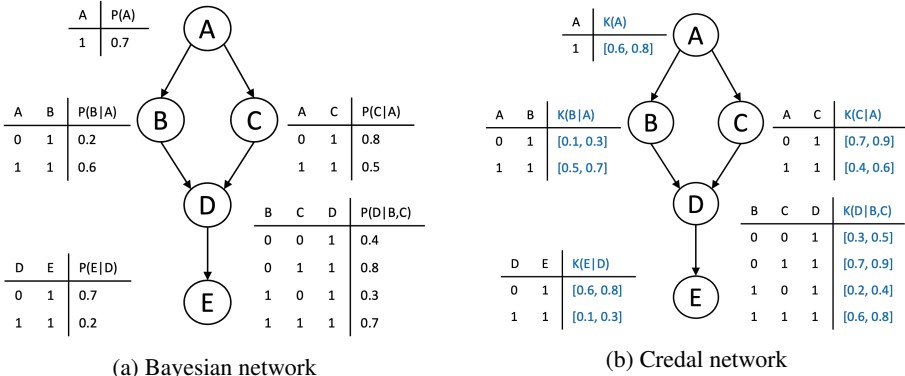

(a) Bayesian network

(b) Credal network

Figure 1: Examples of Bayesian and credal networks with bi-valued variables.

$\mathbf{X}$ as nodes, and $\mathbf{K} = \{K(X_i|\Pi_i = \pi_{ik})\}$ is a set of separately specified conditional credal sets for each variable $X_i$ and each configuration $\pi_{ik}$ of its parents $\Pi_i$ in $G$. The *strong extension* $K(\mathbf{X})$ of a credal network is the *convex hull* (denoted CH) of all joint distributions that satisfy the following Markov property: every variable is strongly independent of its non-descendants conditional on its parents [8] (see also [8] for more details on strong conditional independence).

$$K(\mathbf{X}) = CH\{P(\mathbf{X}) \ : \ P(\mathbf{X}) = \prod_{i=1}^{n} P(X_i|\Pi_i), P(X_i|\Pi_i = \pi_{ik}) \text{ is a vertex of } K(X_i|\Pi_i = \pi_{ik})\}$$

(2)

**Example 1.** *Figure 1a shows a simple Bayesian network with 5 bi-valued variables $\{A, B, C, D, E\}$. The conditional probability tables are shown next to the nodes. For example, we have that $P(B = 1|A = 0) = 0.2$ and $P(B = 1|A = 1) = 0.6$, respectively. In Figure 1b we show a credal network defined over the same set of variables. In this case, the conditional credal sets associated with the variables are given by closed probability intervals such as, for example, $0.1 \leq P(B = 1|A = 0) \leq 0.3$ and $0.5 \leq P(B = 1|A = 1) \leq 0.7$, respectively.*

Unlike in Bayesian networks, a MAP assignment in a credal network may correspond to more than one marginal distribution. Therefore, we define the following two *Credal Marginal MAP* (CMMAP) tasks:

**Definition 1** (maximin). *Let $\mathcal{C} = \langle \mathbf{X}, \mathbf{D}, \mathbf{K}, G \rangle$ be a credal network whose variables are partitioned into MAP variables $\mathbf{X}_M$ and sum variables $\mathbf{X}_S = \mathbf{X} \setminus \mathbf{X}_M$. The maximin Credal Marginal MAP task is finding the assignment $\mathbf{x}_M^*$ to $\mathbf{X}_M$ with the maximum lower marginal probability, namely:*

$$\mathbf{x}_M^* = \underset{\mathbf{X}_M}{\operatorname{argmax}} \min_{P(\mathbf{X}) \in K(\mathbf{X})} \sum_{\mathbf{X}_S} \prod_{i=1}^{n} P(X_i|\Pi_i)$$

(3)

**Definition 2** (maximax). *Let $\mathcal{C} = \langle \mathbf{X}, \mathbf{D}, \mathbf{K}, G \rangle$ be a credal network whose variables are partitioned into MAP variables $\mathbf{X}_M$ and sum variables $\mathbf{X}_S = \mathbf{X} \setminus \mathbf{X}_M$. The maximax Credal Marginal MAP task is finding the assignment $\mathbf{x}_M^*$ to $\mathbf{X}_M$ with the maximum upper marginal probability, namely:*

$$\mathbf{x}_M^* = \underset{\mathbf{X}_M}{\operatorname{argmax}} \max_{P(\mathbf{X}) \in K(\mathbf{X})} \sum_{\mathbf{X}_S} \prod_{i=1}^{n} P(X_i|\Pi_i)$$

(4)

Solving CMMAP can be shown to be $\text{NP}^{\text{NP}^{\text{PP}}}$-hard [15, 11]. Despite being much harder than MMAP, CMMAP is applicable for explaining evidence in imprecise probabilistic models [12, 16].

## 3 Related Work

An exact solution to the Bayesian MMAP task given by Equation (1) can be obtained by using the *variable elimination* (VE) algorithm, a form of dynamic programming which eliminates the variables

**Algorithm 1** Variable Elimination for Credal Marginal MAP

---

1: **procedure** CVE$(\mathcal{C}, \mathbf{X}_M, \mathbf{X}_S)$
2:  initialize $\Gamma \leftarrow \emptyset$
3:  **for all** variable $X_i \in \mathbf{X}$ **do**
4:   $\phi = \{p : p \in ext(K(X_i|\Pi_i))\}$
5:   update $\Gamma = \Gamma \cup \{\phi\}$
6:  create constrained elimination ordering $o$
7:  **for all** variable $X_i \in o$ **do**
8:   $\Gamma_{X_i} = \{\phi : \phi \in \Gamma, X_i \in vars(\phi)\}$
9:   update $\Gamma = \Gamma \setminus \Gamma_{X_i}$
10:  **for all** variable $X_i \in o$ **do**
11:   **if** $X_i \in \mathbf{X}_S$ **then**
12:    $\psi = \max\left(\sum_{X_i} \prod\{\phi \in \Gamma_{X_i}\}\right)$
13:   **else**
14:    $\psi = \max\left(\max_{X_i} \prod\{\phi \in \Gamma_{X_i}\}\right)$
15:   let $Y \in vars(\psi)$ be the closest to $X_i$
16:   update $\Gamma_Y = \Gamma_Y \cup \{\psi\}$
17:  initialize $\mathbf{x}_M^* = \emptyset$
18:  **for all** variable $X_i \in reversed(o)$ **do**
19:   **if** $X_i \in \mathbf{X}_M$ **then**
20:    $x_i^* = \text{argmax}_{X_i} \prod\{\phi(\mathbf{x}_M^*) \in \Gamma_{X_i}\}$
21:    $\mathbf{x}_M^* = \mathbf{x}_M^* \cup \{X_i = x_i^*\}$
22:  **return** $\mathbf{x}_M^*$

---

along a constrained elimination order such that all sum variables are eliminated before the MAP variables [2]. The optimal configuration $\mathbf{x}_M^*$ is obtained by a backward pass that proceeds from the last to the first MAP variable in the constrained elimination ordering and processes each MAP variable by an argmax operator conditioned on the previously instantiated MAP variables.

Alternatively, a depth-first branch and bound search can also be conducted to solve the MMAP exactly, guided by an unconstrained join-tree based upper bound which can be re-evaluated fully [17] or incrementally [18] at each step during search. More recently, a collection of exact schemes sensitive to the problem structure have emerged including depth-first branch and bound search, best-first search, memory efficient recursive best-first search as well as anytime weighted best-first search algorithms that traverse an AND/OR search space associated with the MMAP task [4, 19, 20, 21]. These algorithms are guided by an effective weighted mini-bucket partitioning-based heuristic function and are currently the state-of-the-art for exact MMAP inference.

Several approximation schemes for Bayesian MMAP inference, including mini-bucket partitioning, message-passing and variational methods, have been introduced over the years [22, 23, 24, 25, 26]. These methods, however, do not guarantee eventual optimality of their solutions without significantly increasing their memory requirements.

Stochastic local search algorithms have also been developed to approximate MMAP efficiently [3] while the algorithm introduced by [27] that propagates sets of messages between the clusters of a join-tree and the hybrid schemes combining depth-first and best-first AND/OR search [20] provide lower and upper bounds on the optimal MMAP value in an anytime manner.

## 4 Exact Credal Marginal MAP

In this section, we describe two exact algorithms for solving the Credal Marginal MAP tasks defined by Equations (2) and (3), respectively. Due to space limitation, we discuss only the *maximax* case (the *maximin* is analogous). Specifically, we present a variable elimination scheme that eliminates the variables either by summation or by maximization, as well as a depth-first search scheme that finds the optimal MAP assignment by traversing the search space defined by the MAP variables.

### 4.1 Variable Elimination

Algorithm 1 describes our variable elimination procedure for CMMAP which extends the exact method developed previously for marginal inference tasks [9] and operates on *potentials*.

**Definition 3** (potential). *Given a set of variables $\mathbf{Y}$, a* potential *$\phi(\mathbf{Y})$ is a set of non-negative real-valued functions $p(\mathbf{Y})$ on $\mathbf{Y}$. The product of two potentials $\phi(\mathbf{Y})$ and $\psi(\mathbf{Z})$ is defined by $\phi(\mathbf{Y}) \cdot \psi(\mathbf{Z}) = \{p \cdot q : p \in \phi, q \in \psi\}$. The sum-marginal $\sum_{\mathbf{Z}} \phi(\mathbf{Y})$ and the max-marginal $\max_{\mathbf{Z}} \phi(\mathbf{Y})$ of a potential $\phi(\mathbf{Y})$ with respect to a subset of variables $\mathbf{Z} \subseteq \mathbf{Y}$ are defined by $\sum_{\mathbf{Z}} \phi(\mathbf{Y}) = \{\sum_{\mathbf{Z}} p(\mathbf{Y}) : p \in \phi\}$ and $\max_{\mathbf{Z}} \phi(\mathbf{Y}) = \{\max_{\mathbf{Z}} p(\mathbf{Y}) : p \in \phi\}$, respectively.*

Since the multiplication operator may grow the size of potentials dramatically, we introduce an additional pruning operation that can reduces the cardinality of a potential. Specifically, the operator $\max \phi(\mathbf{Y})$ returns the set of non-zero maximal elements of $\phi(\mathbf{Y})$, under the partial order $\geq$ defined component-wise as $p \geq q$ iff $\forall \mathbf{y}_k \in \mathbf{D_Y}, p(\mathbf{y}_k) \geq q(\mathbf{y}_k)$, where $\mathbf{D_Y}$ is the cartesian product of the domains of the variables in $\mathbf{Y}$: $\max \phi(\mathbf{Y}) = \{p \in \phi(\mathbf{Y}) : \nexists q \in \phi, q \geq p\}$.

Given a credal network $\mathcal{C} = \langle \mathbf{X}, \mathbf{D}, \mathbf{K}, G \rangle$ as input together with a partitioning of its variables into disjoint subsets $\mathbf{X}_M$ (as MAP variables) and $\mathbf{X}_S$ (as sum variables), algorithm CVE transforms each conditional credal set $K(X_i|\Pi_i)$ into a corresponding potential that contains the set of all conditional probability distributions in the strong extension of $K(X_i|\Pi_i)$ (lines 3–5). Subsequently, given an ordering $o$ of the variables in which all the MAP variables come after the sum variables, the potentials are partitioned into buckets. A bucket is associated with a single variable $X_i$ and contains every unallocated potential $\phi$ that has $X_i$ in its scope $vars(\phi)$ (lines 6–9). The algorithm then processes each bucket, from first to last in the constrained elimination ordering $o$, by multiplying all potentials in the current bucket and eliminating the bucket's variable (by summation for sum variables, and by maximization for MAP variables), resulting in a new potential which is first pruned by its non-maximal elements and then placed in a subsequent bucket, depending on its scope (lines 10–16). Following the top-down elimination phase, a bottom-up pass over the MAP buckets, from the last to the first MAP variable in the ordering, assembles the solution $\mathbf{x}_M^*$ by selecting the value $x_i^*$ of variable $X_i$ that maximizes the combination of potentials in its bucket, conditioned on the already assigned MAP variables in the ordering (lines 18–21). Note that the bucket $X_i$'s combined potential may contain more than one components. In this case, we choose the value $x_i^*$ that maximizes the largest number of components in that potential (breaking ties arbitrarily). Clearly, we have the following complexity result:

**Theorem 1** (complexity). *Given a credal network $\mathcal{C}$, the complexity of algorithm CVE is time and space $O(n \cdot C \cdot k^{w_o^*})$, where $n$ is the number of variables, $k$ bounds the domain sizes, $w_o^*$ is the induced width of the constrained elimination order $o$ and $C$ bounds the cardinality of the potentials.*

## 4.2 Depth-First Search

An alternative approach to solving CMMAP exactly is to conduct a depth-first search over the space of partial assignments to the MAP variables, and, for each complete MAP assignment $\mathbf{x}_M$ compute its score as the exact upper probability $\overline{P}(\mathbf{x}_M)$. This way, the optimal solution $\mathbf{x}_M^*$ corresponds to the configuration with the highest score. Evaluating $\overline{P}(\mathbf{x}_M)$ can be done by using a simple modification of the CVE algorithm described in the previous section. Specifically, given a complete assignment $\mathbf{x}_M$ to the MAP variables, the modified CVE, denoted by CVE$^+$, computes an *unconstrained* elimination ordering of all the variables regardless of whether they are MAP or summation variables. Then, for each MAP variable $X_i$ and corresponding value $x_i \in \mathbf{x}_M$, CVE$^+$ adds to the bucket of $X_i$ a deterministic potential $\phi(X_i) = \{\delta_{x_i}\}$, where $\delta_{x_i}$ returns one if $X_i = x_i$ and zero otherwise. Finally, CVE$^+$ eliminates all variables by summation and obtains the desired upper probability bound after processing the bucket of the last variable in the ordering.

**Theorem 2** (complexity). *Given a credal network $\mathcal{C}$, the complexity of the depth-first search algorithm is time $O(n \cdot C \cdot k^{m+w_u^*})$ and space $O(n \cdot C \cdot k^{w_u^*})$, where $n$ is the number of variables, $m$ is the number of MAP variables, $k$ is the maximum domain size, $w_u^*$ is the induced width of the unconstrained elimination ordering $u$ and $C$ bounds the cardinality of the potentials.*

# 5 Approximate Credal Marginal MAP

Solving the CMMAP task exactly is computationally hard and does not scale to large problems. Therefore, in this section, we present several approximation schemes using the mini-bucket partitioning as well as stochastic local search combined with approximate credal marginal inference.

## 5.1 Mini-Buckets Approximation

The first approximation scheme is described by Algorithm 2 and adapts the mini-bucket partitioning scheme developed for graphical models [28] to the CMMAP task. Specifically, algorithm CMBE($i$) is parameterized by an i-bound $i$ and works by partitioning large buckets into smaller subsets, called *mini-buckets*, each containing at most $i$ distinct variables (line 5). The mini-buckets are processed

---

**Algorithm 2** Mini-Buckets for Credal Marginal MAP

---

1: **procedure** CMBE($\mathcal{C}$, $\mathbf{X}_M$, $\mathbf{X}_S$, i-bound)
2:   create constrained elimination ordering $o$
3:   initialize buckets $\Gamma_{X_i}$ as in Algorithm 1
4:   **for all** variable $X_i \in o$ **do**
5:     create mini-buckets $\{Q_1, \ldots, Q_l\}$ of $\Gamma_{X_i}$
6:     **for all** mini-bucket $Q_j, j \in \{1, \ldots, l\}$ **do**
7:       **if** $X_i \in \mathbf{X}_S$ **then**

8:         $\psi = \max \sum_{X_i} \prod\{\phi \in Q_j\}$
9:       **else**
10:        $\psi = \max \max_{X_i} \prod\{\phi \in Q_j\}$
11:       let $Y \in vars(\psi)$ be the closest to $X_i$
12:       update $\Gamma_Y = \Gamma_Y \cup \{\psi\}$
13:   generate $\mathbf{x}_M^*$ as in Algorithm 1
14:   **return** $\mathbf{x}_M^*$

---

separately, as follows: MAP mini-buckets (in $\mathbf{X}_M$) are eliminated by maximization, while variables in $\mathbf{X}_S$ are eliminated by summation. In practice, however, for variables in $\mathbf{X}_S$, one (arbitrarily selected) is eliminated by summation, while the rest of the mini-buckets are processed by maximization. Clearly, CMBE($i$) outputs an upper bound on the optimal maximax CMMAP value from Equation 4.

**Theorem 3** (complexity). *Given a credal network $\mathcal{C}$, the complexity of algorithm CMBE($i$) is time and space $O(n \cdot C \cdot k^i)$, where $n$ is the number of variables, $k$ is the maximum domain size, $i$ is the mini-bucket i-bound and $C$ bounds the cardinality of the potentials.*

## 5.2 Local Search

The second approximation scheme is described by Algorithm 3 and combines stochastic local search with approximate marginal inference for credal networks [29, 30]. More specifically, the basic idea behind the method is to start from an initial guess $\mathbf{x}_M$ as a solution, and iteratively try to improve it by moving to a better neighbor $\mathbf{x}_M'$ that has a higher score. A *neighbor* $\mathbf{x}_M'$ of instantiation $\mathbf{x}_M$ is defined as an instantiation $\mathbf{x}_M'$ which results from changing the value of a single variable $X$ in $\mathbf{x}_M$. For example, the neighbors of $\mathbf{x}_M : (B = 0, C = 1, D = 0)$ for the credal network from Figure 1b are $(B = 1, C = 1, D = 0)$, $(B = 0, C = 0, D = 0)$ and $(B = 0, C = 1, D = 1)$, respectively. In this case, computing the score $score(\mathbf{x}_M')$ of a neighbor $\mathbf{x}_M'$ requires estimating the upper probability of the evidence $\overline{P}(\mathbf{x}_M')$ represented by the assignment $\mathbf{x}_M'$. This can be done efficiently using any of the approximation schemes developed for marginal inference in credal networks such as L2U [29], GL2U [30] or ApproxLP [31]. However, since these schemes were originally designed to compute the lower and upper marginal probabilities of a query variable $Z = z$ conditioned on evidence $\mathbf{Y} = \mathbf{y}$, we use a simple transformation of the credal network to evaluate the probability of evidence $P(\mathbf{Y} = \mathbf{y})$ (see the supplementary material for more details). We present next three strategies for conducting the local search for CMMAP.

**Stochastic Hill Climbing.** Firstly, procedure SHC in Algorithm 3 describes our Stochastic Hill Climbing based approach for CMMAP. Specifically, SHC proceeds by repeatedly either changing the state of the variable that creates the maximum score change (line 13), or changing a variable at random (lines 9 and 15). The quality of the solution returned by the method depends to a large extent on which part of the search space it is given to explore. Therefore, our scheme restarts the search from a different initial solution which is initialized uniformly at random (lines 3-4).

**Taboo Search.** Secondly, procedure TS in Algorithm 3 implements the Taboo Search approach for CMMAP. Taboo search is similar to stochastic hill climbing except that the next neighbor of the current solution is chosen as the best neighbor that hasn't been visited recently. A taboo list maintains a portion of the previously visited solutions so that at the next step a unique point is selected. Our TS algorithm implements a random restarts strategy.

**Simulated Annealing.** Finally, procedure SA in Algorithm 3 describes our Simulated Annealing based scheme for CMMAP. The basic principle behind this approach is to consider some neighboring state $\mathbf{x}_M'$ of the current state $\mathbf{x}_M$, and probabilistically decides between moving to state $\mathbf{x}_M'$ or staying in the current state. The probability of making the transition from $\mathbf{x}_M$ to $\mathbf{x}_M'$ is specified by an acceptance probability function $P(\mathbf{x}_M', \mathbf{x}_M, T)$ that depends on the scores of the two states as well as a global time-varying parameter $T$ called *temperature*. We chose $P(\mathbf{x}_M', \mathbf{x}_M, T) = e^{\frac{\Delta}{T}}$,

**Algorithm 3** Local Search for Credal Marginal MAP

1: **procedure** SHC($\mathcal{C}$, $\mathbf{X}_M \subseteq \mathbf{X}$, $p_{flip}$)
2:   initialize $\mathbf{x}_M^* \leftarrow \emptyset$, $best \leftarrow -\infty$
3:   **for all** iterations $i = 1 \ldots N$ **do**
4:    initialize $\mathbf{x}_M$ randomly
5:    **for all** flips $j = 1 \ldots M$ **do**
6:     sample randomly $p \in (0, 1)$
7:     let $\mathcal{N}$ be $\mathbf{x}_M$'s neighbors
8:     **if** ($p \leq p_{flip}$) **then**
9:      select random neighbor $\mathbf{x}_M' \in \mathcal{N}$
10:     **else**
11:      **for all** neighbor $\mathbf{x}_M' \in \mathcal{N}$ **do**
12:       compute $score(\mathbf{x}_M')$
13:      select highest score neighbor $\mathbf{x}_M'' \in \mathcal{N}$
14:      **if** $score(\mathbf{x}_M'') \leq score(\mathbf{x}_M)$ **then**
15:       select random neighbor $\mathbf{x}_M' \in \mathcal{N}$
16:      **else**
17:       select $\mathbf{x}_M' \leftarrow \mathbf{x}_M''$
18:     **if** $score(\mathbf{x}_M') > best$ **then**
19:      $\mathbf{x}_M^* \leftarrow \mathbf{x}_M'$
20:      $best \leftarrow score(\mathbf{x}_M')$
21:     $\mathbf{x}_M \leftarrow \mathbf{x}_M'$
22:   **return** $\mathbf{x}_M^*$
23: **procedure** TS($\mathcal{C}$, $\mathbf{X}_M \subseteq \mathbf{X}$)
24:   initialize $\mathbf{x}_M = \emptyset$, $best \leftarrow -\infty$
25:   **for all** iterations $i = 1 \ldots N$ **do**
26:    initialize $\mathbf{x}_M$ randomly
27:    $\mathcal{T} \leftarrow \emptyset$
28:    **for all** flips $j = 1 \ldots M$ **do**
29:     $\mathcal{T} \leftarrow \mathcal{T} \cup \{\mathbf{x}_M\}$
30:     let $\mathcal{N}$ be $\mathbf{x}_M$'s neighbors
31:     initialize $\mathbf{x}_M'' \leftarrow \emptyset$, $b \leftarrow -\infty$
32:     **for all** neighbor $\mathbf{x}_M' \in \mathcal{N}$ **do**
33:      **if** $\mathbf{x}_M' \notin \mathcal{T}$ and $score(\mathbf{x}_M') > b$ **then**
34:       $\mathbf{x}_M'' \leftarrow \mathbf{x}_M'$

35:      $b \leftarrow score(\mathbf{x}_M')$
36:     **if** $\mathbf{x}_M'' = \emptyset$ **then**
37:      select random neighbor $\mathbf{x}_M' \in \mathcal{N}$
38:     **else**
39:      $\mathbf{x}_M' \leftarrow \mathbf{x}_M''$
40:     **if** $score(\mathbf{x}_M') > best$ **then**
41:      $\mathbf{x}_M^* \leftarrow \mathbf{x}_M'$
42:      $best \leftarrow score(\mathbf{x}_M')$
43:     $\mathbf{x}_M \leftarrow \mathbf{x}_M'$
44:     **if** $size(\mathcal{T}) \geq S$ **then**
45:      prune $\mathcal{T}$ until $size(\mathcal{T}) < S$
46:   **return** $\mathbf{x}_M^*$
47: **procedure** SA($\mathcal{C}$, $\mathbf{X}_M \subseteq \mathbf{X}$, $T_{init}$, $\sigma$)
48:   initialize $\mathbf{x}_M^*$ randomly
49:   $best \leftarrow score(\mathbf{x}_M^*)$
50:   **for all** iterations $i = 1 \ldots N$ **do**
51:    set $\mathbf{x}_M \leftarrow \mathbf{x}_M^*$, $T \leftarrow T_{init}$
52:    **for all** flips $j = 1 \ldots M$ **do**
53:     let $\mathcal{N}$ be $\mathbf{x}_M$'s neighbors
54:     select random neighbor $\mathbf{x}_M' \in \mathcal{N}$
55:     $\Delta \leftarrow \log score(\mathbf{x}_M') - \log score(\mathbf{x}_M)$
56:     **if** $\Delta > 0$ **then**
57:      $\mathbf{x}_M \leftarrow \mathbf{x}_M'$
58:     **else**
59:      sample randomly $p \in (0, 1)$
60:      **if** $p < e^{\frac{\Delta}{T}}$ **then**
61:       $\mathbf{x}_M \leftarrow \mathbf{x}_M'$
62:     **if** $score(\mathbf{x}_M) > best$ **then**
63:      $\mathbf{x}_M^* \leftarrow \mathbf{x}_M$
64:      $best \leftarrow score(\mathbf{x}_M)$
65:    $T \leftarrow T * \sigma$
66:   **return** $\mathbf{x}_M^*$

where $\Delta = \log \overline{P}(\mathbf{x}_M') - \log \overline{P}(\mathbf{x}_M)$. At each iteration, the temperature is decreased using a cooling schedule $\sigma < 1$. Like SHC and TS, algorithm SA implements a random restarts strategy.

**Theorem 4** (complexity). *Given a credal network $\mathcal{C}$, the complexity of algorithms SHC, TS and SA is time $O(N \cdot M \cdot P)$ and space $O(n)$, where $n$ is the number of variables, $N$ is the number of iterations, $M$ is the maximum number of flips allowed per iteration, and $P$ bounds the complexity of approximating the probability of evidence in $\mathcal{C}$.*

## 6 Experiments

We evaluate the proposed algorithms for CMMAP on random credal networks and credal networks derived from real-world applications. All competing algorithms were implemented in C++ and the experiments were run on a 32-core machine with 128GB of RAM running Ubuntu Linux 20.04.

We consider the two exact algorithms denoted by CVE and DFS, as well as the four approximation schemes denoted by SHC, TS, SA and CMBE($i$), respectively. The local search algorithms used $N = 10$ iterations and $M = 10,000$ maximum flips per iteration, and they all used the approximate L2U algorithm with 10 iterations [29] to evaluate the MAP assignments during search. Furthermore, for SHC we set the flip probability $p_{flip}$ to 0.2, TS used a taboo list of size 100, while for SA we set

| $n$ | Q | $w^*$ | SHC time (#) | W | TS time (#) | W | SA time (#) | W | CMBE(2) time (#) | W |
|---|---|---|---|---|---|---|---|---|---|---|
| | | | | | | random | | | | |
| | 20 | 25 | 32.69 | 100 | 23.08 | 100 | **6.47** | 100 | 225.28 (70) | 1 |
| 100 | 40 | 37 | 163.05 | 100 | 79.11 | 100 | **14.78** | 100 | 327.67 (43) | 0 |
| | 60 | 23 | 421.93 | 100 | 185.41 | 100 | **29.99** | 100 | 224.93 (7) | 0 |
| | 30 | 39 | 254.32 | 100 | 141.03 | 100 | **24.08** | 100 | 294.48 (43) | 0 |
| 150 | 60 | 57 | 1143.78 | 100 | 531.45 | 100 | **70.06** | 100 | 555.98 (14) | 0 |
| | 90 | 66 | 2811.47 | 100 | 1259.79 | 100 | **139.78** | 75 | 925.41 (2) | 0 |
| | 50 | 58 | 1044.79 | 100 | 490.38 | 100 | **72.09** | 100 | 276.98 (38) | 0 |
| 200 | 100 | 86 | 3496.77 (32) | 32 | 2143.79 | 100 | **211.47** | 14 | 927.31 (1) | 0 |
| | 150 | 69 | 3601.67 (16) | 16 | 3550.99 (17) | 17 | **339.34** | 72 | - (0) | 0 |
| | | | | | | grid | | | | |
| | 20 | 25 | 31.35 | 100 | 22.77 | 100 | 4.66 | 100 | **0.07** | 2 |
| 100 | 40 | 37 | 155.34 | 100 | 79.83 | 100 | 10.51 | 100 | **3.85** | 0 |
| | 60 | 23 | 358.81 | 100 | 168.79 | 100 | 19.18 | 100 | **28.76** | 0 |
| | 30 | 36 | 219.49 | 100 | 121.86 | 100 | 21.02 | 100 | **0.34** | 0 |
| 144 | 60 | 53 | 878.63 | 100 | 426.70 | 100 | 54.77 | 100 | **1.03** | 0 |
| | 90 | 26 | 2109.47 | 100 | 958.13 | 100 | 102.93 | 73 | **27.79** | 0 |
| | 50 | 55 | 817.52 | 100 | 382.46 | 100 | 58.13 | 100 | **0.68** | 0 |
| 196 | 100 | 56 | 3045.54 (94) | 94 | 1453.39 | 100 | 147.11 | 13 | **51.01** (98) | 0 |
| | 150 | 22 | 3601.25 (23) | 23 | 3011.47 (93) | 93 | 190.26 | 3 | **41.27** (99) | 2 |
| | | | | | | $k$-tree | | | | |
| | 20 | 25 | 68.25 | 100 | 44.25 | 100 | **10.48** | 100 | 221.18 (55) | 0 |
| 100 | 40 | 37 | 307.91 | 100 | 151.97 | 100 | **23.19** | 100 | 163.59 (8) | 0 |
| | 60 | 23 | 650.72 | 100 | 306.26 | 100 | **40.19** | 100 | - (0) | 0 |
| | 30 | 28 | 443.33 | 100 | 245.71 | 100 | **44.58** | 100 | 492.55 (26) | 0 |
| 150 | 60 | 47 | 1647.29 | 100 | 724.01 | 100 | **106.71** | 100 | 14.68 (1) | 0 |
| | 90 | 51 | 2917.01 (84) | 84 | 1541.91 | 100 | **192.76** | 82 | - (0) | 0 |
| | 50 | 45 | 1306.43 | 100 | 660.59 | 100 | **108.36** | 100 | 1199.83 | 0 |
| 200 | 100 | 64 | 3376.59 (54) | 54 | 1917.95 | 100 | **266.24** | 21 | - (0) | 0 |
| | 1500 | 48 | 3602.98 (4) | 4 | 3334.49 (59) | 59 | **344.96** | 38 | - (0) | 0 |

Table 1: Results on `random`, `grid` and $k$-tree credal networks. Mean CPU times in seconds, number of instance solved (#) and number of wins (W). Time limit 1 hour, 8GB of RAM.

the initial temperature and cooling schedule to $T_{init} = 100$ and $\sigma = 0.9$, respectively. For CMBE($i$) we set the i-bound $i$ to 2 and used the same L2U algorithm to evaluate the solution found. All competing algorithms were allocated a 1 hour time limit and 8GB of memory per problem instance.

In all our experiments, we report the CPU time in seconds, the number of problems solved within the time/memory limit and the number of times an algorithm converged to the best possible solution. The latter is called the number of *wins* and is meant to be a measure of solution quality for the respective algorithm. We also record the number of variables ($n$), the number (or percentage) of MAP variables (Q) and the constrained induced widths ($w^*$). The best performance points are highlighted.

## 6.1 Random Credal Networks

For our purpose, we generated random credal networks, $m$-by-$m$ grid networks as well as $k$-tree networks. Specifically, for the random networks, we varied the number of variables $n \in \{100, 150, 200\}$, for grids, we choose $m \in \{10, 14, 16\}$, and for $k$-trees we selected $k = 2$ and the number of variables $n \in \{100, 150, 200\}$, respectively. In all cases, the maximum domain size was set to 2 and the conditional credal sets were generated uniformly at random as probability intervals such that the difference between the lower and upper probability bounds was at most 0.3.

First, we note that the exact algorithms CVE and DFS could only solve very small problems with up to 10 variables and 5 MAP variables. The main reason for the poor performance of these algorithms is the extremely large size of the intermediate potentials generated during the variable elimination procedure which causes the algorithms to run out of memory or time on larger problems. Therefore, we omit their evaluation hereafter.

| problem | $w^*$ | SHC time (#) | W | TS time (#) | W | SA time (#) | W | CMBE(2) time (#) | W |
|---|---|---|---|---|---|---|---|---|---|
| alarm | 12 | 27.32 (10) | 10 | 21.31 (10) | 10 | **4.89** (10) | 10 | 324.23 (10) | 0 |
| child | 7 | 3.51 (10) | 10 | 3.69 (10) | 10 | 1.19 (10) | 10 | **0.64** (10) | 0 |
| link | 239 | 3655.67 (2) | 2 | 3628.15 (2) | 1 | **1300.14** (10) | 8 | - (0) | 0 |
| insurance | 12 | 64.22 (10) | 10 | 45.77 (10) | 10 | **17.11** (10) | 10 | 97.16 (9) | 0 |
| hepar2 | 25 | 1734.11 (10) | 10 | 833.82 (10) | 10 | **163.16** (10) | 10 | - (0) | 0 |
| pathfinder | 33 | 2509.89 (10) | 10 | 79.78 (10) | 10 | **93.98** (10) | 10 | - (0) | 0 |
| hailfinder | 14 | 126.60 (10) | 10 | 72.73 (10) | 10 | **12.53** (10) | 10 | 531.52 (10) | 0 |
| largefam | 402 | - (0) | 0 | - (0) | 0 | **2903.55** (10) | 10 | - (0) | 0 |
| mastermind1 | 389 | - (0) | 0 | - (0) | 0 | **3600.85** (10) | 10 | - (0) | 0 |
| mastermind2 | 726 | - (0) | 0 | - (0) | 0 | **3617.17** (5) | 5 | - (0) | 0 |
| mastermind3 | 1193 | - (0) | 0 | - (0) | 0 | **3650.54** (3) | 3 | - (0) | 0 |
| mildew | 12 | 22.49 (10) | 10 | 15.22 (10) | 10 | 3.15 (10) | 10 | **0.16** (10) | 0 |
| munin | 175 | 3615.45 (5) | 4 | 3639.57 (3) | 3 | **652.72** (10) | 5 | - (0) | 0 |
| pedigree1 | 74 | 3603.87 (7) | 7 | 3609.44 (8) | 8 | **410.61** (10) | 0 | 1269.45 (8) | 0 |
| pedigree7 | 147 | 3620.74 (2) | 1 | 3625.51 (2) | 1 | **1689.89** (10) | 9 | - (0) | 0 |
| pedigree9 | 175 | - (0) | 0 | - (0) | 0 | 1719.92 (10) | 6 | **1128.50** (4) | 4 |
| win95pts | 28 | 3612.38 (6) | 6 | 3610.94 (6) | 6 | **821.20** (10) | 10 | - (0) | 0 |
| xandes | 75 | 3619.00 (6) | 6 | 3611.76 (6) | 6 | **3565.62** (10) | 3 | - (0) | 0 |
| xdiabetes | 75 | 3605.08 (9) | 9 | 3603.76 (10) | 10 | **175.76** (10) | 0 | 182.38 (6) | 0 |
| zbarley | 18 | 140.93 (10) | 10 | 82.36 (10) | 10 | **18.47** (10) | 10 | 863.64 (1) | 0 |
| zpigs | 105 | 3606.83 (4) | 4 | 3603.8 (5) | 5 | 234.84 (10) | 5 | **100.47** (2) | 0 |
| zwater | 16 | 207.07 (10) | 10 | 126.87 (10) | 10 | **44.68** (10) | 10 | - (0) | 0 |

Table 2: Results on real-world credal networks with $Q = 50\%$ MAP variables. Mean CPU time in seconds, number of instances solved (#) and number of wins (W). Time limit 1 hour, 8 GB of RAM.

Table 1 summarizes the results obtained on random, grid and $k$-tree networks. Each data point represents an average over 100 random problem instances generated for each problem size ($n$) and number of MAP variables (Q), respectively. Next to the running time we show the number of instances solved within the time/memory limit (if the number is omitted then all 100 instances were solved). We can see that in terms of running time, CMBE(2) performs best on the grid networks. This is because the intermediate potentials generated during elimination are relatively small size and therefore are processed quickly. However, the algorithm is not able converge to good quality solutions compared with its competitors. The picture is reversed on the random and $k$-tree networks where CMBE(2) is the worst performing algorithm both in terms of running time and solution quality. In this case, the relatively large intermediate potentials cause the algorithm to exceed the time and memory limits on many problem instances and thus impact negatively its performance.

The local search algorithms SHC, TS and SA yield the best performance in terms of solution quality with all three algorithms almost always converging to the best possible solutions on these problem instances. In terms of running time, SA is the fastest algorithm achieving almost one order of magnitude speedup over its competitors, especially for larger numbers of MAP variables (e.g., $k$-trees with $n = 100$ variables and $Q = 60$ MAP variables). Algorithms SHC and TS have comparable running times (with SHC being slightly slower than TS) but they are significantly slower than SA. This is due to the significantly larger computational overhead required for evaluating the scores of all the neighbors of the current state, especially when there are many MAP variables.

## 6.2 Real-World Credal Networks

Table 2 shows the results obtained on a set of credal networks derived from 22 real-world Bayesian networks[1] by converting the probability values in the CPTs into probability intervals such that the difference between the corresponding lower and upper probability bounds was at most 0.3. Furthermore, since the local search algorithms rely on the L2U approximation to evaluate the MAP configurations, we restricted the domains of the multi-valued variables to the first two values in the domain while shrinking and re-normalizing the corresponding CPTs. For each network we selected uniformly at random $Q = 50\%$ of the variables to act as MAP variable and generated 10 random instances. As before, we indicate next to the average running times the number of instances solved

---

[1] Available at https://www.bnlearn.com/bnrepository/

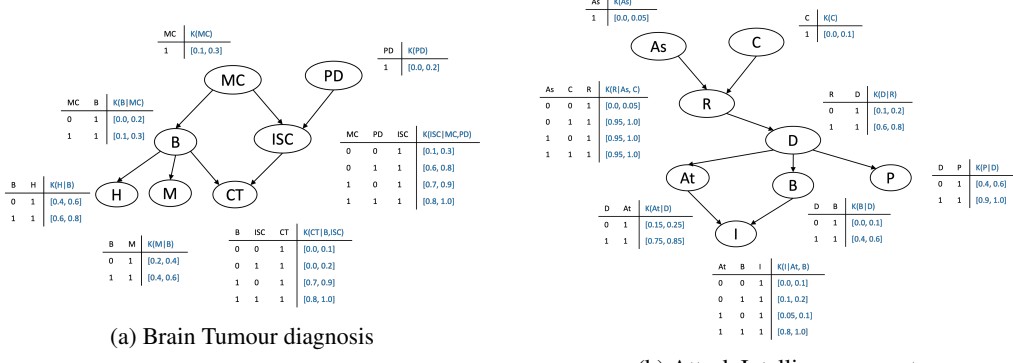

(a) Brain Tumour diagnosis

(b) Attack Intelligence report

Figure 2: Applications of Credal Marginal MAP inference.

by the respective algorithms within the time and memory limits. We can see again that CMBE(2) is competitive only on the easiest instances (e.g., child, mildew) while SA yields the best performance in terms of both running time and solution quality on the majority of the problem instances. In summary, the relatively large potentials hinder CMBE's performance, while the computational overhead incurred during the evaluation of relatively large neighborhoods of the current state slows down significantly SHC and TS compared with SA.

### 6.3 Applications

Figure 2a shows the credal network for the brain tumour diagnosis use case derived from the Bayesian network described in [11]. The variables are: MC - metastatic cancer, PD - Paget disease, B - brain tumour, ISC - increased serum calcium, H - headaches, M - memory loss, CT - scan result.

Considering the query variables $B$ and $ISC$, the exact solution for both maximax and maximin CMMAP is $(B = 0, ISC = 0)$ (obtained by both the CVE and DFS algorithms). In this case, the maximax and maximin scores are $0.837$ and $0.42$, respectively. Algorithms SHC, TS and SA also find the optimal configuration $(B = 0, ISC = 0)$ which is evaluated by L2U to $0.8316$ for maximax CMMAP and to $0.37296$ for maximin CMMAP, respectively.

Figure 2b shows the credal network for the intelligence report analysis described in [9]. The variables are: As - assassination, C - coup/revolt, R - regime change, D - decision to invade, At - attack, B - build-up, P - propaganda, I - invasion.

Considering the query variables $D$, $At$ and $I$, the exact solution for both maximax and maximin CMMAP is $(D = 0, At = 0, I = 0)$ and is obtained by both algorithms CVE and DFS. The corresponding scores are in this case $0.765$ and $0.458806$, respectively. The approximation schemes SHC, TS and SA also find the same optimal CMMAP configuration $(D = 0, At = 0, I = 0)$ which is evaluated by L2U to $0.69651$ for maximax CMMAP and to $0.305486$ for maximin CMMAP, respectively.

We note that in both cases, the constrained induced width is 2 and therefore CMBE(2) coincides with the exact CVE. Therefore, all our approximation schemes found the optimal solutions.

## 7 Conclusions

The paper explores the Marginal MAP inference task in credal networks. We formally define the Credal Marginal MAP task and present new exact algorithms based on variable elimination and depth-first search. Subsequently, we introduce approximate algorithms using the mini-bucket partitioning or a combination of stochastic local search and approximate credal marginal inference. Our experiments on random and real-world credal networks demonstrate the effectiveness of our CMMAP algorithms. A potential direction for future work is to investigate branch-and-bound and best-first search strategies guided by a credal version of weighted mini-buckets [32].

**Acknowledgements**

Fabio Cozman was supported in part by C4AI funding (FAPESP and IBM Corporation, grant 2019/07665-4) and CNPq (grant 305753/2022-3).

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
