# OpenReview forum: "Credal Marginal MAP"
_NeurIPS.cc/2023/Conference — NeurIPS 2023 poster_

### Official Review · Reviewer_8qY7 · 2023-07-03

**Soundness:** 3 good
**Presentation:** 3 good
**Contribution:** 3 good
**Rating:** 7
**Confidence:** 4

**Summary:**

The paper studies algorithms for the marginal MAP (MMAP) problem in Credal networks that generalize Bayesian networks. The paper gives an overview of the problem, existing algorithms for Bayesian networks and generalizes the algorithms to credal networks. Overall, two exact and multiple heuristic approaches are presented. As the exact approaches were unable to deal with larger problems, the experiments are restricted to heuristic approaches.

**Strengths:**

The paper is very well written and gives a good overview of the problem and related work. While Bayesian networks are an old topic, they remain relevant for probabilistic reasoning tasks that require analytical guarantees. However, one problem of Bayesian networks is that specifying the CPDs can be difficult. Credal networks are an interesting generalization that allows using probability intervals rather than point probabilities in order to capture the uncertainty about the encoded knowledge. To the best of my knowledge, there is not much literature on algorithms for Credal networks. The authors generalize state-of-the-art ideas for Bayesian networks to Credal networks and give an empirical evaluation. The paper is therefore an interesting contribution to the probabilistic reasoning literature. The code is attached in the supplementary material to reproduce the experimental results and is a useful resource for the probabilistic reasoning community.

**Weaknesses:**

It could be discussed in more detail what the exact relationship between the proposed algorithms and the corresponding algorithms for BNs is. Are they generalizations in the sense that if the intervals in the Credal network are tight, then the Credal network algorithms correspond to the BN algorithms? Are there challenges in generalizing the algorithms to credal nets or is the generalization straightforward? And how do the runtime guarantees for Credal networks compare to those for Bayesian networks?

**Questions:**

Could you elaborate on the relationship between CN and BN algorithms (see weaknesses)?

**Limitations:**

Yes

---

> ### Author Rebuttal · Authors · 2023-08-07
>
> Thank you for your comments and suggestions.
>
> Regarding the relationship between the CN and BN algorithms, in principle, if the probability intervals in a CN are tight, namely they collapse to point probabilities, then the CMMAP task we defined for CNs collapses to the MMAP task for BNs. In this case, the CMBE algorithm, for example, will be identical to the MBE algorithm for MMAP in BNs because CMBE will no longer have to propagate sets of potentials.
>
> However, in the more general case when we introduce probability intervals and/or more complex credal sets, the two tasks are different from a computational complexity point of view: MMAP for BNs is NP^PP-complete, whereas MMAP for CNs is NP^NP^PP-complete. Therefore, the corresponding MMAP algorithms will be significantly different (i.e., those for CNs vs those for BNs).

---

> > ### Comment · Reviewer_8qY7 · 2023-08-16
> >
> > Thank you for the explanations.

---

### Official Review · Reviewer_VJAx · 2023-07-04

**Soundness:** 3 good
**Presentation:** 3 good
**Contribution:** 3 good
**Rating:** 6
**Confidence:** 4

**Summary:**

This paper is about a generalisation of marginal MAP (MMAP) for Bayesian networks (BNs). The authors allow the BN parameters to vary in (credal ) sets. The goal is, therefore, to find the configuration with the maximum upper (wrt the credal sets) probability. The authors first consider exact inference. A number of schemes based on mini-buckets are then obtained to address approximate inference and empirically validated.

**Strengths:**

The problem is important and very general (in a sense, most of the classical inferences in PGMs can be intended as a subcase of CMMMAP). This seems to be the first serious attempt to address the problem in the credal case and the experiments show how the proposed approximate schemes allows to solve a large number of non-trivial instances.

**Weaknesses:**

The authors only consider maxi-max and maxi-min versions of the problem. Credal networks are often used to model a condition of indecision between multiple options, and considering other decision criteria possibly leading to multiple options (e.g., maximality or interval dominance) would be interesting. This is also related to the ideas sketched by De Bock et al. (Neurips 2014).


**Questions:**

The ApproxLP scheme based on linearisation is used to address MAR in CNs. Can we extend it to MMAP and then perform a comparison against the methods proposed here? This point is partially addressed in Section 5.2, but the answer seems to be a bit inconclusive.

CMBE is giving an upper bound. Do the search methods give a lower bound? If so, is it never the case that the two results coincide, this meaning that the solution is exact?

**Limitations:**

I don't see significant sociated impacts for the present work.

---

> ### Author Rebuttal · Authors · 2023-08-07
>
> Thank you for your comments and suggestions.
>
> We are already exploring several ideas around interval dominance as well as alternative criteria such as maximality and/or E-admissibility. Therefore, we are very thankful for your suggestions and we plan to address these issues in our future work.
>
> Extending ApproxLP to CMMAP is indeed an interesting idea. We believe that in principle it is possible to do so. Therefore, it is an excellent suggestion of future work together with developing new, more efficient approximation schemes for CMMAP.
>
> At the moment, the search based schemes can give a lower bound only if the approximate marginal inference algorithm used to evaluate the current solution is guaranteed to provide a lower bound on the probability of evidence (either the lower probability or the upper probability). And yes, in principle, if we have that the upper bound produced by CMBE is the same as the lower bound produced by another scheme then they both found the exact solution. Therefore, one direction of future work is to develop a scheme that computes in an anytime manner both lower and upper bounds on the CMMAP values.

---

> > ### Comment · Reviewer_VJAx · 2023-08-21
> >
> > Thanks for the clarifications. I am happy to confirm my positive opinion about the paper.

---

### Official Review · Reviewer_3qEE · 2023-07-05

**Soundness:** 4 excellent
**Presentation:** 3 good
**Contribution:** 2 fair
**Rating:** 5
**Confidence:** 3

**Summary:**

This work presents novel algorithms for performing exact and approximate marginal MAP inference in credal networks with discrete-valued factors, and evaluates the computational and inferential effectiveness of these algorithms on a number of benchmarks.

**Strengths:**

Firstly, I enjoyed reading the paper and think it’s fantastic how basic research into PGM inference is still being done. A strength of the paper is the technical depth demonstrated to devise these novel inference algorithms. I thought the design of the ablation study was solid and plenty of real-world experiments given.

**Weaknesses:**

One weakness is that the explanation of the algorithms proposed is difficult to follow, but perhaps this is unavoidable with a heavily technical topic. I think it would strengthen the paper to explain the significance of marginal MAP in credal networks in particular, and performing experiments on counterfactual analysis. More discussion/experiments on counterfactual inference may strengthen the apparent contribution/impact of solving MMAP in credal networks.

**Questions:**

Is the case where the credal network has arbitrary real-valued distributed variables, rather than all discrete-valued variables, important for any of the applications discussed?

**Limitations:**

Method seems only to apply to Bayesian Networks in the narrow sense of the term with discrete factors.

---

> ### Author Rebuttal · Authors · 2023-08-07
>
> Thank you for your comments and suggestions.
>
> We will revise the presentation and will try to expand the discussion of the algorithms in order to address the concerns identified during the review. Certainly the content is very technical but we will do our best to make it more didactic.
>
> We appreciate your suggestion to perform experiments on counterfactual analysis. We will definitely take it into consideration and try to formalize the connection between the CMMAP task and counterfactual analysis. We emphasize that causal reasoning is a clear application of credal networks (as explained in our Ref. [12]), a point that, we think, adds to the relevance of the paper to NeurIPS. More specifically, the estimation of distributions for exogenous variables from the observation of endogenous variables directly leads to credal networks, so our techniques can be directly applied to counterfactual reasoning given endogenous distributions and associated directed graphs that capture causal mechanisms (as in fact shown by the paper by Zaffalon et al).
>
> Regarding your questions about real-valued variables, yes, we believe that it is almost always the case that in real-world applications many variables of interest are real-valued. Therefore, we will adopt the broad definition of Bayesian/credal networks as the focus on discrete factors is a feature of this paper but a larger perspective including continuous variables can be very important in applied work and may be a path for future research.
>
> Furthermore, real-valued variables typically require developing different kinds of algorithms compared with the variable elimination and search based ones we proposed in this paper. Handling real-valued variables is also an important direction of our future work.

---

### Official Review · Reviewer_9XGw · 2023-07-06

**Soundness:** 4 excellent
**Presentation:** 4 excellent
**Contribution:** 3 good
**Rating:** 7
**Confidence:** 4

**Summary:**

The paper proposes inference algorithms for credal networks for the marginal MAP inference task. The idea is to use variable elimination methods for this task. An exact inference algorithm is proposed as well as approximations using mini-bucket partitioning. Further, stochastic local search procedures combined with existing approximate marginal inference methods are proposed to solve the MMAP task. Experiments are performed on randomly generated credal networks as well as real-world Bayesian networks converted to credal networks.


**Strengths:**

This looks to be the first work on MMAP for credal networks which seems significant since MMAP is a hard but important task for PGMs. The use of existing inference algorithms with the local search methods gives a general family of MMAP algorithms. The evaluation considers a large number of benchmarks. The paper is generally well-written with clear contributions.

**Weaknesses:**

The relatively poor performance of CMBE may indicate it is hard to get more reliable performance for MMAP (e.g. by increasing i-bound of CMBE) for credal networks compared to approximations for other PGMs. The stochastic local search algorithms which seem to work much better in the experiments may be harder to trade-off w.r.t accuracy vs complexity.

**Questions:**

What strategies could we use to control/trade-off accuracy vs complexity in the algorithms where the results show best performance (e.g. SA)?

**Limitations:**

There are no limitations explicitly mentioned.

---

> ### Author Rebuttal · Authors · 2023-08-07
>
> Thank you for your comments and suggestions.
>
> One way to address the tradeoff accuracy vs complexity in the case of algorithm CMBE for example is to also bound the size of the sets of potentials that are propagated during elimination (or approximate somehow these sets of potentials). This way we may be able to increase the i-bound (thus hoping to improve accuracy) while lowering to some extent the computational complexity of the elimination process. Clearly, we will need to expand the empirical evaluation of this proposed scheme; we hope our paper will open an interesting line of research exploring better bounds for this provably hard problem.
>
> For the local search based algorithm we may be able to lower the computational effort  by developing a faster, incremental scheme to compute the scores of all neighbours using the score of the current solution. At the moment, each neighbour’s score is computed from scratch using an approximate marginal inference algorithm and this negatively impacts the running time especially on larger problem instances. The comments by the reviewer do suggest exciting paths to be pursued.

---

> > ### Comment · Reviewer_9XGw · 2023-08-18
> > **Thanks**
> >
> > Thanks for your response and look forward to follow-on improvements to this work.

---

### Decision · Program_Chairs · 2023-09-21

**Decision:**

Accept (poster)

**Comment:**

This paper studies marginal MAP inference, which is already notoriously hard on graphical models. It does so on credal graphical models, where accounting for the parameter uncertaintly makes the problem even more challenging. The paper is novel in studying this problem and the empirical evaluation is thorough.